# A Double-Blind, Randomized, Placebo-Controlled Trial of the Effect of 1-Kestose on Defecation Habits in Constipated Kindergarten Children: A Pilot Study

**DOI:** 10.3390/nu15143276

**Published:** 2023-07-24

**Authors:** Mayuko Takahashi, Yoshihiro Kadota, Yuki Shiko, Yohei Kawasaki, Kenichi Sakurai, Chisato Mori, Naoki Shimojo

**Affiliations:** 1B Food Science Co., Ltd., Chita 478-0046, Aichi, Japan; m-takahashi@bfsci.co.jp (M.T.); y-kadota@bfsci.co.jp (Y.K.); 2Clinical Research Center, Chiba University Hospital, Chiba 260-8677, Chiba, Japan; shiko_yuki@chiba-u.jp; 3Japanese Red Cross College of Nursing, Shibuya 150-0012, Tokyo, Japan; ykawasaki@chiba-u.jp; 4Center for Preventive Medical Sciences, Chiba University, Chiba 260-8670, Chiba, Japan; sakuraik@faculty.chiba-u.jp (K.S.); cmori@faculty.chiba-u.jp (C.M.)

**Keywords:** constipation, prebiotic, 1-kestose, *Intestinibacter*, gut microbiota, short-chain fatty acid

## Abstract

Constipation is common in children and can significantly affect quality of life. Prebiotics are reportedly helpful for constipation in adults, but few studies have examined their use in young children. In this study, the effect of 1-kestose (kestose), which has excellent bifidobacterial growth properties, on constipation in kindergarten children (*n* = 11) was compared with that of maltose (*n* = 12) in a randomized, double-blind study. Three grams of kestose per day for 8 weeks did not affect stool properties, but significantly increased the number of defecations per week (Median; 3 → 4 times/week, *p* = 0.017, effect size = 0.53). A significant decrease in *Intestinibacter*, a trend toward increased bifidobacteria, and a trend toward decreased *Clostridium sensu stricto* were observed after kestose ingestion, while concentrations of short-chain fatty acids in stools were unchanged.

## 1. Introduction

Many children experience chronic constipation, including mild cases. Constipation in children is primarily attributed to diet, incorrect defecation habits, and lifestyle factors. Without appropriate treatment, a vicious cycle of constipation may develop and persist into adulthood [1]. The standard treatment for constipation is long-term laxative therapy with lifestyle modification. However, patients with relatively mild pathology do not require medical attention or medication, and there is no established treatment for these patients. One option other than pharmacotherapy is to improve the intestinal microbiota [2].

Probiotics are often used to improve the intestinal microbiota [3]. Compared with probiotics, prebiotics have the advantages of acting on the bacterial flora originally present in the host, being effective against a wider range of bacteria, and being easier to take. Several reports have shown that prebiotics improve constipation in adults [4]. On the other hand, few studies have examined the effect of prebiotics on defecation habits in children. A 6-week study of inulin in 2- to 5-year-old children with constipation [5] and a 4-week study of a dietary fiber mixture in 4- to 12-year-old children [6] mainly evaluated clinical parameters such as improvements in stool characteristics and frequency of bowel movements, but did not study the actual gut microbiota.

1-Kestose (kestose) is a fructooligosaccharide (Appendix A), a type of prebiotic material, and is known to have a higher growth potential for beneficial intestinal bacteria such as bifidobacteria, lactic acid bacteria, and butyrate-producing bacteria in vitro compared with other prebiotics [7,8,9]. The present study was designed to evaluate the clinical efficacy of kestose on improving bowel movements in kindergarten children in relation to intestinal microbiota and metabolites.

## 2. Materials and Methods

### 2.1. Clinical Study Design

We conducted a randomized, double-blind, placebo-controlled study of children attending one of two kindergartens in Chiba City from September 2021 to December 2021. The kindergartners themselves and their parents were asked about their willingness to participate in the study, and those who expressed a willingness to participate were asked to complete a preliminary survey. In the pre-survey, participants record their bowel movements for one week. Inclusion criteria were kindergartners who reported defecation on ≤4 days per week in the pre-survey. Exclusion criteria were (1) those with a chronic disease or a history of serious disease, (2) those with allergies to the study foods, and (3) those who planned to start or finish taking prebiotic or probiotic preparations or foods during the study period. Eligible subjects were randomly assigned to either a kestose or a maltose group. The allocation factor was facility (per kindergarten), with stratified randomization and assignment to two groups. The allocation was performed by a different person than the one conducting the study and data analysis. Because this was a pilot study to estimate the effect of kestose on constipated infants, no effect size was obtained a priori. Therefore, the number of cases was designed based on feasibility.

Participants consumed 3 g of kestose (Kestose 95; B Food Science Co., Aichi, Japan) or 3 g of maltose (Sunmalt; Hayashibara Co., Okayama, Japan) once a day for 8 weeks. Maltose was chosen as a placebo with no significant effect on the gut microbiota. Daily defecation was recorded for 16 weeks, and the frequency of defecation, volume of defecation, fecal characteristics, and color of feces were investigated. Defecation volume was visually assessed as the approximate equivalent of the number of chicken eggs (ranging from 1/4 to 10 eggs). Stool characteristics were rated on a 7-point Bristol scale [10]. The color of stools was evaluated on a 6-point scale according to color intensity.

Stool samples were also collected from participants on the start date and after 8 weeks and 16 weeks of test-food consumption. Samples were temporarily stored at −20 °C, then at −80 °C until used for analysis.

The test schedule is shown in Appendix A.

### 2.2. Ethical Considerations

This study was conducted in accordance with the Declaration of Helsinki for experiments involving human participants. This study was approved by the ethics committee of the Graduate School of Medicine at Chiba University (approval no. #4027) and registered in UMIN-CTR (UMIN000044252). Written informed consent was obtained from the guardians of all subjects after providing an explanation about the study to the subjects and their guardians.

### 2.3. DNA Extraction

Bacterial DNA was extracted from fecal samples as described by Takahashi et al. [11]. Briefly, frozen fecal samples (0.1 g) were added to a tube that contained 4 M guanidium thiocyanate, 100 mM Tris-HCl (pH 9.0), and 40 mM ethylenediaminetetraacetic acid (EDTA). Samples were then homogenized with zirconium beads using a Precellys Evolution instrument (Bertin Instruments, Montigny-le-Bretonneux, France). DNA was extracted from the homogenized suspensions using a Magtration System 12GC and GC series MagDEA DNA 200 (Precision System Science, Chiba, Japan). DNA was purified using an Automated DNA extraction system PI-480 and NR-201 (Kurabo Industries, Osaka, Japan) DNA concentrations were estimated using spectrophotometry (NanoDrop ND8000; Thermo Fisher Scientific, Waltham, MA, USA).

### 2.4. 16S rRNA Gene-Sequence Analysis Using Next-Generation Sequencing (NGS)

Fecal bacterial 16S rRNA gene (16S rDNA) was analyzed with NGS using an MiSeq system (Illumina, San Diego, CA, USA), as previously described [11]. The V3-V4 hypervariable regions of 16S rDNA were amplified using polymerase chain reaction from microbial genomic DNA using the universal primers for bacteria (341f and R806) and the dual-index method [12]. Barcoded amplicons were sequenced using the paired-end method and were modified to a 2 × 284-bp cycle run on the MiSeq system using MiSeq Reagent Kit version 3 (600 Cycle) (Illumina). After alignment, overlapping regions within the paired-end reads were merged and primer regions were omitted, resulting in a 430 bp sequence. Only reads with ≥99% of the sequence with quality value scores ≥20 were extracted for further analysis [11]. The chimeric sequence detected by Usearch6.1.544_i86 was excluded [13]. Based on the resulting sequences, taxonomic positions of the sequences were identified at 97% similarity using Metagenome@KIN analysis software Ver.2.21(World Fusion, Tokyo, Japan) and the TechnoSuruga Lab Microbial Identification database DB-BA 13.0 (TechnoSuruga Laboratory, Shizuoka, Japan) [12,14].

### 2.5. Measurement of Short-Chain Fatty Acids (SCFAs)

The measurement of SCFAs, including acetate, propionate, butyrate, isobutyrate, valerate, and isovalerate, was performed via Gas Chromatography—Mass spectrometry (GC/MS) (Shimazu, Kyoto, Japan) on Rtx-1701 columns (Restec, Bellefonte, PA, USA). GC/MS samples were prepared as follows: 50 mg (wet weight) of fecal sample was suspended in 300 µL of pure water. The suspension was centrifuged at 15,000× *g* at 4 °C for 5 min. A total of 20 µL of 20 mM 2-ethyl butyrate, 80 µL of 3 N HCl, and 1000 µL of diethylether were then added to the supernatant. This was thoroughly stirred and the supernatant was obtained via centrifugation to make the SCFA solution sample.

### 2.6. Statistical Analysis

Statistical analyses were performed using SPSS Statistics 26 software package (SPSS Inc., Chicago, IL, USA) and Microsoft Excel (Excel version in Microsoft Office 2016 for Windows, Microsoft, WA, USA). The normality of data was examined using the Shapiro–Wilk test. Corresponding two-group comparisons were made using the Wilcoxon signed-rank test. For comparisons between two groups without correspondence, the Mann–Whitney U test was used. Pearson’s correlation analysis was used for correlation analysis. Differences were considered significant at the level of *p* < 0.05.

## 3. Results

### 3.1. Participant Characteristics

A pre-survey of 67 preschool children was conducted to assess eligibility. Twenty-five children met the inclusion criteria with defecation on ≤4 days/week, but two withdrew consent to participate, so twenty-three participants were investigated. Participants were randomly assigned to the kestose group (*n* = 11) and maltose group (*n* = 12).

One participant in the kestose group who developed soft stools during the study period was withdrawn from the study. Statistical analysis was performed on the 10 remaining participants in the kestose group and 12 participants in the maltose group (Appendix A).

No significant differences in age, sex, or use of pre/probiotics or laxatives were identified between groups (Appendix A).

### 3.2. Defecation Diary

Defecation results before and after 8 weeks are shown in Figure 1. No significant differences were observed between groups at baseline. In the kestose group, significant increases were observed in the number of defecations per week, number of days with defecation per week, and total volume of stool per week. In contrast, the maltose group showed a significant increase in median weekly stool volume. After 16 weeks, no significant differences from week 0 were identified for any items in both groups. No change over time was observed in stool characteristics or stool color.

### 3.3. 16S rRNA Gene Metagenomic Analysis of Intestinal Microbiota before and after Kestose Ingestion

The 16S rRNA metagenomic analysis was performed for the 0- and 8-week stool samples from the kestose group and for the 0-week sample from the maltose group. Mean (±standard error) total numbers of reads analyzed at 0 and 8 weeks from the kestose group and 0 weeks from the maltose group were 28,416 ± 937, 29,984 ± 1210, and 28,655 ± 633, respectively. Shannon’s alpha diversity index (25–75%) calculated from the number of species reads was 2.60 (2.40–2.86), 2.71 (2.57–2.89), and 2.67 (2.49–2.89), respectively. No significant differences were identified in either case. Relative abundance ratios of each bacterial genus, calculated from the number of reads, are shown for the 16 bacterial genera that had a median relative abundance ratio of at least 0.5% at either time point. Group comparisons of occupancy between the kestose and maltose groups at 0 weeks showed no significant differences in any bacterial genera. Results for the 0- and 8-week samples from the kestose group are shown in Table 1. The most dominant genus at both time points was *Bifidobacterium* spp. *Intestinibacter* spp. showed a significant decrease, *Bifidobacterium* spp. showed an increasing trend, and *Clostridium sensu stricto* showed a decreasing trend.

### 3.4. Concentrations of SCFAs in Stool in the Kestose Group

Concentrations of SCFAs (acetate, propionate, butyrate, isobutyrate, valerate, and isovalerate) were measured from the 0- and 8-week stool samples for the kestose group. Appendix A shows concentrations of three SCFAs (acetate, propionate, butyrate) in stool. At all times, acetic acid had the highest concentration, followed by propionic acid and butyric acid. Although SCFA concentrations were higher at 8 weeks compared with 0 weeks, no significant changes were observed over time in the kestose group.

Total SCFA levels (sum of acetate, propionate, and butyrate) did not change significantly between weeks 0 and 8, and the percentage of each SCFA to total SCFAs was not observed to change over time.

## 4. Discussion

In the present randomized, double-blind, placebo-controlled trial investigating 23 children who initially had defecation on ≤4 days/week, we observed that kestose intake improved defecation habits by increasing the number of bowel movements per week, number of days with bowel movements, and total number of bowel movements. Median weekly stool volume was also increased in the maltose (placebo) group, but total stool volume was not increased, suggesting that defecation was unimproved. Intervention studies on the effect of prebiotics on constipation in children are scarce, with very few data showing efficacy [15], but a meta-analysis in adults has shown efficacy [16]. Although the study was conducted with younger subjects, infants treated with galactooligosaccharides (GOS) also showed a significant increase in stool frequency compared with the control group [17]. These results were compatible with the results of this study, which showed that prebiotic intake increased the frequency of bowel movements.

In the kestose group, there was no change in the alpha diversity index of the intestinal microflora and a trend toward increased *Bifidobacterium*, similar to previous reports of changes in microflora due to kestose intake [18,19]. Many studies of inulin treatment in constipated adults have reported an increase in *Bifidobacterium*, with increased relative abundance of *Anaerostipes*, *Faecalibacterium*, and *Lactobatillus* as a common result in some studies [20]. Similarly, in the present study, *Bifidobacterium* was increased, but no significant increase was observed for the other species. This may be due to differences in intervention targets. In addition, the relative abundance of *Intestinibacter* was decreased. *Intestinibacter* has not been reported to be associated with constipation, but belongs to *Clostridium* cluster XI, which has been reported as harmful [21], and the number of bacteria in *Clostridium* cluster XI has been negatively correlated with the Bristol Stool Characteristics Scale [22]. The reduction in relative abundance of *Intestinibacter* was similar to previous results [23,24], suggesting a change to a better intestinal environment following kestose ingestion. *Clostridium* sensu strico also showed a decreasing trend. It has been reported that the presence of *Clostridium* sensu strico during infancy is associated with the development of atopic dermatitis [25]. It is possible that *Bifidobacterium* and *Clostridium* sensu stricto did not confirm statistically significant differences due to the insufficient number of cases. The results of this study reflected the improvement of the intestinal microflora due to kestose intake.

Since SCFAs have been shown to promote colon peristalsis and defecation [26], we measured SCFAs in stools. As mentioned above, an increase in defecation frequency was observed with kestose intake, which predicted an increase in short-chain fatty acid production, but contrary to our expectations, no significant changes in stool concentrations of SCFAs were seen between before and after kestose intake. A recent study by Shibata et al. regarding the effects of kestose on cow’s milk allergies in children suggested that kestose decreased the milk-specific immunoglobulin E levels in serum, but SCFAs in stools were unchanged [19]. Our findings appear compatible with that observation and do not support the involvement of SCFAs in any improvement of bowel movements following kestose intake in children. However, the present study measuring stool concentrations was likely unable to detect changes over time, since kestose are rapidly fermented in the ascending colon [27] and the majority of SCFAs produced in the intestinal tract are rapidly absorbed [28]. Other trials in which prebiotics were administered to patients with constipation also showed improvement in constipation, but the levels of short-chain fatty acids in the stool were not significantly different [29]. It is possible that this is the reason why this study, like previous studies, showed no change in the concentration of short-chain fatty acids in stools, despite changes in the relative abundance of intestinal microflora.

The limitations of this study include the small number of subjects studied, partly because subjects in this study were selected through a preliminary survey. A larger study is needed to include analyses of bacterial flora and intestinal metabolites. Using an effect size of 0.53 for this study, with a power of 0.8, α = 0.05, and a corresponding *t*-test, 30 cases are needed.

In conclusion, kestose intake appeared to improve defecation and reduced the relative abundance of *Intestinibacter* among preschool children with constipation.

## 5. Conclusions

Kestose increased the number of bowel movements per week, the number of days with bowel movements, and the total amount of bowel movements per week compared with maltose. It also decreased the relative abundance of *Intestinibacter* spp. in stools, but no correlation with defecation results was observed. The concentration of short-chain fatty acids in the stool did not change before or after ingestion and did not correlate with the results of defecation or fecal flora.

## Figures and Tables

**Figure 1 nutrients-15-03276-f001:**
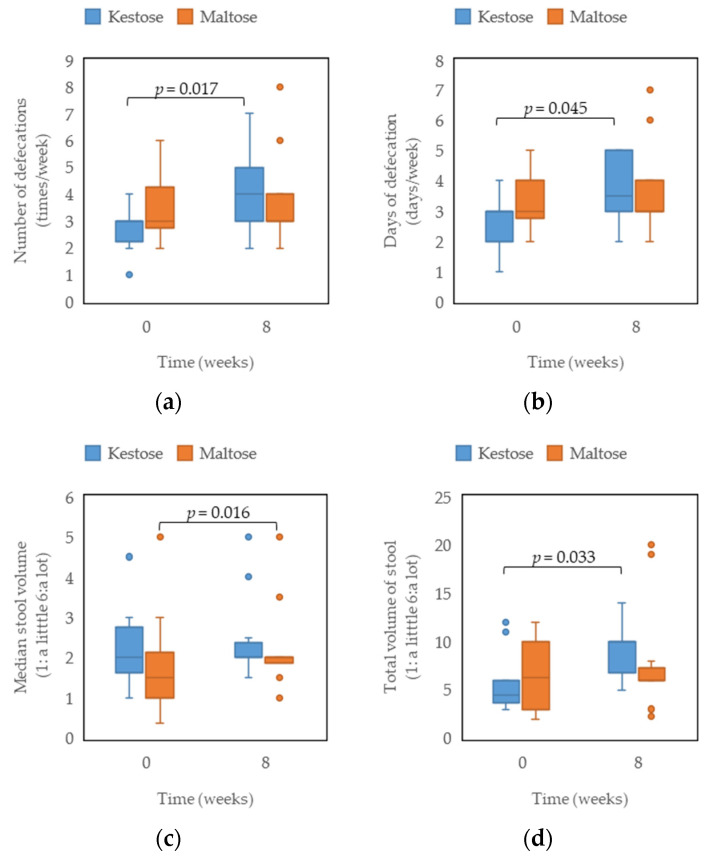
Summary of participants’ defecation habits at the beginning and end of the study. Color points are outliers (values greater than 1.5 times the interquartile range). (**a**) Total weekly number of defecations (minimum: 0 times); (**b**) number of days with defecation in a week (minimum: 0 days, maximum: 7 days); (**c**) median weekly stool volume (1 = size of chicken egg); (**d**) total stool volume per week.

**Table 1 nutrients-15-03276-t001:** Relative abundance ratios of gut microbiota at the genus level in the kestose group before and after intervention.

	Relative Abundance Median (25th–75th %)		*p*
	Kestose (*n* = 10)		
	0 Weeks	8 Weeks	
*Bifidobacterium*	22.2 (13.3–32.2)	29.2 (17.2–42.9)	0.074
*Blautia*	14.7 (8.7–18.0)	13.1 (10.4–15.3)	0.445
*Fusicatenibacter*	4.5 (2.3–6.3)	5.4 (4.5–8.2)	0.721
*Anaerostipes*	4.4 (2.6–6.7)	3.0 (1.7–5.6)	0.959
*Collinsella*	3.9 (0.2–6.3)	4.7 (0.3–9.3)	0.446
*Gemmiger*	3.6 (0.4–5.1)	3.1 (0.4–5.4)	0.953
*Anaerobutyricum*	3.2 (0.7–4.6)	2.8 (0.3–3.6)	0.646
*Streptococcus*	2.5 (2.1–5.4)	1.6 (1.0–7.3)	0.878
*Ruminococcus*	1.7 (0.3–3.6)	1.5 (0.3–2.5)	0.333
*Bacteroides*	1.6 (0.5–2.3)	1.1 (0.1–3.6)	0.799
*Agathobacter*	1.6 (0.0–2.4)	0.9 (0.2–1.5)	0.674
*Intestinibacter*	1.3 (0.4–1.9)	0.3 (0.2–0.6)	0.047
*Clostridium sensu stricto*	1.0 (0.3–2.3)	0.5 (0.1–0.7)	0.074
*Romboutsia*	0.9 (0.3–5.1)	1.5 (0.6–3.3)	1.000
*Lachnospiracea_incertae_sedis*	0.7 (0.3–1.9)	0.6 (0.2–1.3)	0.508
*Faecalibacterium*	0.6 (0.2–2.6)	1.0 (0.6–4.2)	0.441

## Data Availability

Not applicable.

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
