# Peer review of "A Double-Blind, Randomized, Placebo-Controlled Trial of the Effect of 1-Kestose on Defecation Habits in Constipated Kindergarten Children: A Pilot Study"

_nutrients, 2023, doi:10.3390/nu15143276_

Round 1

Reviewer 1 Report

This is an interesting study in a topical area. Randomised placebo controlled trials  are  rare and difficult to do in this age group so the authors are to be congratulated on this albeit rather small study.

General Comment

The study is too small to be convincing but I accept recruitment is difficult in this group.

Specific comments

1.       Abstract This should include number of subjects and effect size Please give change in number of defecations and p values

2.       Introduction: You state “   Without appropriate treatment, a vicious cycle of constipation may develop and persist 26 into adulthood.” Although this sounds reasonable do you have any references to support this? I am not aware of any long term studies

3.       “One option other than pharmacotherapy is to improve the intestinal microbiota” this needs supporting reference.

4.       Might be worth having an image showing 1-ketose as some reader may not be familiar with it  and explaining it is a  fructose molecule connected to sucrose by a (1→2β) glycosidic bond

5.       Methods: This needs to give inclusion and exclusion criteria. Surely you exclude those taking antibiotics?

6.       Can you indicate if there were any palatability issues Did any child drop out because of taste?

7.       Was the study preregistered on a clinical trial registry describing its endpoints?

8.       Can you indicate the force of the bead beating ?  This is known to alter yield from firmicutes.

9.       Please clarify how stool volume was assessed and by whom

10.    Can you state the% recovery of spiked samples for the SCFAs to understand

11.    It is important to record exactly how randomisation was done to ensure meeting quality standard for any meta-analysis.

12.    Please indicate how you calculated the power of the study and how you decided on how many participants to recruit.

13.    Results : Not clear why the subject who developed soft stools was withdrawn. Was this patient choice? You ought to include in any intention to treat analysis.

14.    It looks from Table 1 as if prebiotic intake was permitted . Surely this should have been stopped during the trial Can you clarify?

15.    Figure 3 legend needs clarification  What does 1 chicken egg mean? Do you mean 1= size of chicken egg??

16.    Microbiota You need to state that the differences in Bifidobacteria could have been due to chance and then give the p value to allow the reader to decide if it is signicant. In truth it looks as if you rs study was underpowered

17.    Figure 4 is missing on my version Please show

18.    Discussion : You should explain that kestose is rapidly fermented in the ascending colon and any increase in SCFA would have disappeared by the time chyme reached the rectum and was expelled. Fecal SCFAs do not reliably indicate what occurs in the right colon. They tend to increase when transit time is markedly reduced since then absorption of SCFAs is incomplete

19.    You acknowledge the study was underpowered but it would be useful to calculate how many subject you would need to achieve 90% power to detect the difference in Bifidobacteria.

Well written only minor edits needed

Many children experience chronic constipation, including mild cases needs rephrasing

Do you mean

Many children experience chronic constipation if you include mild cases.

Or better

Constipation is common in children with a wide range of severity.

Author Response

Corrections in the text are highlighted in yellow.

Point 1: Abstract This should include number of subjects and effect size Please give change in number of defecations and p values.

Response 1: We have Added on lines 15-18.

Point 2: Introduction: You state “   Without appropriate treatment, a vicious cycle of constipation may develop and persist 26 into adulthood.” Although this sounds reasonable do you have any references to support this? I am not aware of any long term studies

Response 2: Reference added on lines 27.

Point 3: “One option other than pharmacotherapy is to improve the intestinal microbiota” this needs supporting reference.

Response 3: We included a review paper showing probiotics may improve constipation.

Point 4: Might be worth having an image showing 1-ketose as some reader may not be familiar with it  and explaining it is a  fructose molecule connected to sucrose by a (1→2β) glycosidic bond

Response 4: A supplemental figure was added and an explanation was added in the text.

Point 5: Methods: This needs to give inclusion and exclusion criteria. Surely you exclude those taking antibiotics?

Response 5: Added to the text. Treatments including antibiotics were not reported by subjects.

Point 6: Can you indicate if there were any palatability issues Did any child drop out because of taste?

Response 6: No subjects dropped out due to taste quality. Kestose has a sweet taste similar to sugar.

Point 7: Was the study preregistered on a clinical trial registry describing its endpoints?

Response 7: As noted in the text, we registered with UMIN prior to the start of the study.

Point 8: Can you indicate the force of the bead beating ?  This is known to alter yield from firmicutes.

Response 8: The bead beat was performed twice at 5.0 m/s, 60 sec.

Point 9: Please clarify how stool volume was assessed and by whom

Response 9: The subject's parent or guardian looked at the subject's stool and assessed its approximate size.

Point 10: Can you state the% recovery of spiked samples for the SCFAs to understand

Response 10: In this study, 2-ethylbutyrate was used as an internal standard when measuring SCFA. No spike evaluation was performed.

Point 11: It is important to record exactly how randomisation was done to ensure meeting quality standard for any meta-analysis.

Response 11: Allocation factors and allocators were added (lines 56-59).

Point 12: Please indicate how you calculated the power of the study and how you decided on how many participants to recruit.

Response 12: Since this study is a pilot study, sample size calculation based on the effect size was not performed, but the results of the sample size calculation based on the effect size of this study have been added to the discussion section.

Point 13: Results : Not clear why the subject who developed soft stools was withdrawn. Was this patient choice? You ought to include in any intention to treat analysis.

Response : The patient was instructed to discontinue consumption of the test food, including the possibility that the soft stools were caused by the test food. This patient was excluded from this analysis because he consumed very little of the test food during the study period.

Point 14: It looks from Table 1 as if prebiotic intake was permitted . Surely this should have been stopped during the trial Can you clarify?

Response : Prebiotics/probiotics, including yogurt, are very common today. We did not prohibit prebiotics/probiotics during the study period and instructed the participants not to change their habits during the study period.

Point 15: Figure 3 legend needs clarification  What does 1 chicken egg mean? Do you mean 1= size of chicken egg??

Response 15: Added to the text.

Point 16: Microbiota You need to state that the differences in Bifidobacteria could have been due to chance and then give the p value to allow the reader to decide if it is significant. In truth it looks as if you rs study was underpowered

Response 16: Added to legends in Figure 1.

Point 17: Figure 4 is missing on my version Please show

Response 17: Changed the file format of the figure.

Point 18: Discussion : You should explain that kestose is rapidly fermented in the ascending colon and any increase in SCFA would have disappeared by the time chyme reached the rectum and was expelled. Fecal SCFAs do not reliably indicate what occurs in the right colon. They tend to increase when transit time is markedly reduced since then absorption of SCFAs is incomplete

Response 18: Thank you for your input. I have added it to the discussion.

Point 19: You acknowledge the study was underpowered but it would be useful to calculate how many subject you would need to achieve 90% power to detect the difference in Bifidobacteria.

Response 19: Using an effect size of 0.53 for this study, with a power of 0.8, α = 0.05, and a corresponding t-test, 30 cases are needed. Added at the end of the discussion.

Reviewer 2 Report

The authors have discussed the role of the 1-kestose on defecation habits in constipated kindergarten children. I have the following concerns:

1.     The author only wrote about the inclusion criteria in the article. Is there any exclusion criteria, such as whether some children with congenital diseases will be included?

2.     The number of subjects recruited in this article is relatively small, and the reference basis for sample size calculation is not specified in the article. Please provide it.

3.     During the intervention period, researchers did not restrict children from consuming other defecation assisted products. How to exclude the interference of these pre/probiotics or laxatives in the experiment?

4.     Has the diet of the subjects been controlled during the experiment, and how can the impact of abnormal dietary intake on defecation be rule out?

5.     Is there differences in the number of defecations, days of defecations, median weekly stool volume, total stool volume per week, and composition of gut microbiota between groups after intervention?

6.     There are few discussions about gut microbiota on defecation symptoms. It is better to make a correlation analysis on the phenotype of defecation and gut microbiota to further explore the correlation between them. Please discuss direct correlation of bacterial groups in each group and their possible effects on defecation habits.

7.     Please create a biological project with all sequencing information deposited into a suitable database.

8.     The relationship between SCFA and gut microbiota can also be further discussed. The abundance of bacteria that may produce SCFA can be analyzed, so as to discuss the reason why SCFA do not change.

9.     The manuscript should be throughly revised for many minor language mistakes throughout the manuscript.

1.     The manuscript should be throughly revised for many minor language mistakes throughout the manuscript.

Author Response

Corrections in the text are highlighted in greens.

Point 1: The author only wrote about the inclusion criteria in the article. Is there any exclusion criteria, such as whether some children with congenital diseases will be included?

Response 1: We have Added on lines 56-59 (Yellow highlight, as noted by Reviewer1.).

Point 2: The number of subjects recruited in this article is relatively small, and the reference basis for sample size calculation is not specified in the article. Please provide it.

Response 2: We have Added on lines 62-65 (Yellow highlight, as noted by Reviewer1.).

Point 3: During the intervention period, researchers did not restrict children from consuming other defecation assisted products. How to exclude the interference of these pre/probiotics or laxatives in the experiment?

Response 3: Prebiotics/probiotics, including yogurt, are very common today. We did not prohibit prebiotics/probiotics during the study period and instructed the participants not to change their habits (consuming other defecation assisted products) during the study period.

Point 4: Has the diet of the subjects been controlled during the experiment, and how can the impact of abnormal dietary intake on defecation be rule out?

Response 4: Ideally, the diet should be controlled, but it is not practical to record or control the diet during the 12-week study period. They are instructed not to change their eating habits drastically.

Point 5: Is there differences in the number of defecations, days of defecations, median weekly stool volume, total stool volume per week, and composition of gut microbiota between groups after intervention?

Response 5: Added on lines 152-154.

Point 6: There are few discussions about gut microbiota on defecation symptoms. It is better to make a correlation analysis on the phenotype of defecation and gut microbiota to further explore the correlation between them. Please discuss direct correlation of bacterial groups in each group and their possible effects on defecation habits.

Response 6: We have also investigated the relationship between defecation symptoms and intestinal microbiota using methods other than those shown in the text. Since no correlation was demonstrated in any of these methods, only representative results are presented.

Point 7: Please create a biological project with all sequencing information deposited into a suitable database.

Response 7: Since there was no indication in the study plan and the subject consent that the fecal information obtained would be registered in the database, no registration will be conducted. Many small pilot studies don't register gut microbiota to the public data base. Do you suggest we should deposit it to the data base?

Point 8: The relationship between SCFA and gut microbiota can also be further discussed. The abundance of bacteria that may produce SCFA can be analyzed, so as to discuss the reason why SCFA do not change.

Response 8: It is difficult to discuss SCFA concentrations directly from the species of bacteria because the species of bacteria that produce SCFA vary from genus to genus, and each species has a different production capacity. Since 16SrRNA was used to identify the species in this study, it is difficult to classify the species by SCFA-producing genes.

Point 9:  The manuscript should be throughly revised for many minor language mistakes throughout the manuscript.

Response 9: The text has been edited by native English speakers.

Round 2

Reviewer 2 Report

The revision is appropriate.